# uPAR Knockout Results in a Deep Glycolytic and OXPHOS Reprogramming in Melanoma and Colon Carcinoma Cell Lines

**DOI:** 10.3390/cells9020308

**Published:** 2020-01-28

**Authors:** Alessio Biagioni, Anna Laurenzana, Anastasia Chillà, Mario Del Rosso, Elena Andreucci, Martina Poteti, Daniele Bani, Daniele Guasti, Gabriella Fibbi, Francesca Margheri

**Affiliations:** 1Department of Experimental and Clinical Biomedical Sciences, University of Florence, Viale G.B. Morgagni 50, 50134 Firenze, Italy; anna.laurenzana@unifi.it (A.L.); anastasia.chilla@unifi.it (A.C.); mario.delrosso@unifi.it (M.D.R.); e.andreucci@unifi.it (E.A.); martina.poteti@gmail.com (M.P.); gabriella.fibbi@unifi.it (G.F.); fmargheri@unifi.it (F.M.); 2Department of Experimental and Clinical Medicine, University of Florence, Largo Brambilla 3, 50134 Firenze, Italy; daniele.bani@unifi.it (D.B.); daniele.guasti@unifi.it (D.G.)

**Keywords:** colon cancer, CRISPR, gene-editing, melanoma, uPAR

## Abstract

Urokinase Plasminogen Activator (uPA) Receptor (uPAR) is a well-known GPI-anchored three-domain membrane protein with pro-tumor roles largely shown in all the malignant tumors where it is over-expressed. Here we have exploited the Clustered Regularly Interspaced Short Palindromic Repeats (CRISPR)/Cas9 gene knock out approach to investigate its role in the oxidative metabolism in human melanoma and colon cancer as the consequences of its irreversible loss. Knocking out *PLAUR*, a uPAR-encoding gene, in A375p, A375M6 and HCT116, which are two human melanoma and a colon carcinoma, respectively, we have observed an increased number of mitochondria in the two melanoma cell lines, while we evidenced an immature biogenesis of mitochondria in the colon carcinoma culture. Such biological diversity is, however, reflected in a significant enhancement of the mitochondrial spare respiratory capacity, fueled by an increased expression of GLS2, and in a decreased glycolysis paired with an increased secretion of lactate by all uPAR KO cells. We speculated that this discrepancy might be explained by an impaired ratio between LDHA and LDHB.

## 1. Introduction

Since the very first description of the receptor for the urokinase plasminogen activator (uPAR) in 1985 [1], a large amount of works have identified the urokinase plasminogen activator system as the mediator of many aspects of cancer cell malignancy, modulating numerous signaling events, cell survival, invasion and migration, angiogenesis and intra-tumor recruitment of inflammatory cells [2,3,4]. As a biomarker in breast cancer, the Plasminogen Activation system is validated for prognostic use in level-of-evidence-1 studies [5]. uPAR over-expression, even in cancer-associated stromal cells, such as cancer-associated fibroblasts (CAFs), is associated with poor prognosis [6]. The integrin-mediated connections of uPAR with many receptor systems, such as receptor-tyrosine-kinases (EGFR, PDGFR), G-protein-coupled receptors and formyl peptide receptor-type 1 [7,8], confers signaling competence to such receptors even in the absence of the relevant specific ligands [9,10]. The classical uPAR-uPA-Plasminogen cell surface cascade, with subsequent extracellular matrix proteolysis, features one of the most important pathways of tumor cells invasion and endothelial cell migration [3,4,11]. Moreover, the role of uPAR in the amoeboid invasion of cancer cells has been recently demonstrated [12]. Given such an abundance of tasks and critical cross-talks, the attempts to control uPAR-mediated multiple functions poses big theoretical and practical problems, such as the privileged inhibition of invasive features that is likely to leave unaltered the signaling properties of the so-called uPAR interactome [4,8]. Our previous efforts, shared by other groups, have pursued the clearing of uPAR from the cell surface by using anti-uPAR ODN (OligoDeoxyNucleotide) [13,14] and miRNA [15], or exploiting uPAR inactivation by specific cleavage systems such as MMP-12 [16] in vitro and in vivo. As a follow-up of our studies, we have used the CRISPR (Clustered Regularly Interspaced Short Palindromic Repeats)/Cas9 technique to obtain the irreversible clearing of uPAR in A375p and A375M6 human melanoma and in HCT116 colon cancer cell lines. This technology, based on a naturally-occurring system that protects bacteria from viral infections [17], is now widely used to manipulate genes in cells and living organisms [18,19,20]. Here we show that the uPAR KO clones have undergone a deep metabolic reprogramming, since the uPAR clearing in melanoma and colon carcinoma cell lines impairs the glycolytic pathway while enhancing the mitochondrial spare respiratory capacity and altering mitochondria biogenesis.

## 2. Materials and Methods

### 2.1. Cell Lines

Human melanoma A375p (CRL-1619) and colon cancer HCT 116 (CCL-247) cell lines were obtained from American Type Culture Collection while the human melanoma cell line A375-M6 was isolated from lung metastasis of SCID bg/bg mice IV injected with A375p [21] and grown in DMEM with 10% fetal bovine serum (Euroclone, Milano, Italy). Cells were tested every two weeks for Mycoplasma by PCR using two universal primers (MGSO and GPO1) [22].

### 2.2. Transfection and Plasmid

The plasmids (sc-400666-NIC) for CRISPR/Cas9, targeting *PLAUR* exon 3, were obtained from Santa Cruz Biotechnology (Santa Cruz, CA, USA) and transfected according to the manufacturer’s instructions. sgRNA were, however, subjected to off-target sites analysis throughout the Cas-OFF Finder software (http://www.rgenome.net/cas-offinder/). No off-target sites in the loci of the most likely off-target activity of the CRISPR/Cas9 system targeted by the chosen sgRNAs could be detected as shown in Table 1. Cells were then sorted for the Green Fluorescent Protein (GFP) marker and selected with 1 µg/mL puromycin for 2–3 weeks (Sigma-Aldrich, Saint Louis, MO, USA), which is singularly characterized using Western Blotting, qPCR and PCR on the full-length mRNA. For the uPAR rescue expression experiment, cells were stably transfected using an Okayama-Berg vector containing uPAR cDNA, and they were selected with G418 as resistance marker (0.5 mg/mL) as previously reported [23].

### 2.3. Western Blot Analysis

Cell pellets were incubated with a lysis buffer and subjected to Western Blotting as previously described [12]. Primary antibodies used were anti-uPAR (R3) provided by Thermofisher Scientific (Monza, Italy); anti-pAkt, Akt, pERK and ERK provided by Cell Signaling; anti-α-Tubulin provided by Sigma Aldrich and anti-GAPDH by Abcam, Cambridge, UK. Protein bands were analyzed by Odyssey Infrared Imaging System (Lycor Bioscience) using ImageJ software (developed by Wayne Rasband, National Institutes of Health, Bethesda, MD, USA) for protein quantification.

### 2.4. RNA Extraction, Semiquantitative and Quantitative PCR

Total RNA was prepared using Tri Reagent (Sigma-Aldrich), agarose gel-checked for integrity and reverse-transcribed with a cDNA synthesis kit (BioRad, Milano, Italy) according to the manufacturer’s instructions. Selected genes were evaluated by qualitative PCR using Blue Platinum PCR Super Mix (Thermofisher Scientific) or Real-Time RT-PCR using SsoAdvanced Universal Green Mix (BioRad) with a 7500 Fast Real Time PCR System (Applied Biosystems, Waltham, MA, USA). For Real Time RT-PCR, fold change was determined by the comparative Ct method using β2-Microglobulin as the normalization gene. Primer sequences (IDT, TemaRicerca, Bologna, Italy) reported in Table 2 were selected following the previous scientific literature focused on cancer cell metabolism.

### 2.5. Assessment of Metabolic Profile

Respiratory and glycolytic capacity of melanoma and colon cancer cells were performed with the Seahorse XFe96 Analyzer (Seahorse Bioscience, Billerica, MA, USA) on attached cells. Briefly, 2 × 10^4^ cells/well were seeded in quintuplicate in XF96 V7 microplates for 24 h. Before analysis, cells were resuspended in Seahorse assay buffer and the following drugs were added: 1 μM oligomycin, 1 μM FCCP (carbonyl cyanide-4-(trifluoromethoxy)phenylhydrazone), 0.5 μM rotenone/antimycin A for Mito Stress Assay, and 10 mM glucose, 1 μM oligomycin and 50 mM 2-DG for Glycolysis Stress Assay. The results were normalized on the protein content evaluated with the BCA assays. The experiments were performed assaying five technical replicates and at least three biological replicates.

### 2.6. Lactate Assay

Lactate was measured in cultured media with a Lactate Assay kit (BioVision, Vinci-Biochem, Florence, Italy) according to the manufacturer’s instruction.

### 2.7. 2-NBDG Glucose Uptake

The fluorescent-labeled glucose analog, 2-(*N*-(7-nitrobenz-2-oxa-1,3-diazol-4-yl)amino-2-deoxyglucose (2-NBDG; Glucose Uptake Cell-Based Assay Kit, Biovision, Milpitas, CA, USA), was used to measure the glucose uptake by cultured cells, according to the manufacturer’s protocol. For each measurement, data from 10,000 single cell events were collected and analyzed by a flow cytometer.

### 2.8. Flow Cytometry Analysis

Cells were harvested with Accutase (Euroclone), washed once with cold PBS and then stained with fluorochrome-conjugated mAbs anti-CD87 (Thermofisher Scientific) for 1 h on ice in the dark. Cells were analyzed by flow cytometry BD-FACS Canto II with Flowjo Software (BD Biosciences, Milano, Italy).

### 2.9. Transmission Electron Microscope (TEM) Analysis

The cells were collected by Accutase (Euroclone) and centrifuged at 1000 rpm for 5 min in a 1.5 mL Eppendorf tube. The cell pellet was then fixed in isotonic 4% glutaraldehyde and 1% OsO4, dehydrated, and embedded in Epon epoxy resin (Fluka, Buchs, Switzerland) for electron microscopic studies. Ultrathin sections were stained with 1% samarium triacetate-10% gadolinium triacetate (UAR-EMS stain #22405, Electron Microscopy Sciences, Hatfield, PA, USA), counterstained with lead citrate and viewed and photographed under a JEM 1010 transmission electron microscope (Jeol, Tokyo, Japan) equipped with a MegaView III high-resolution digital camera and imaging software (Jeol).

### 2.10. Statistics

Results are expressed as mean ± SD. Multiple comparisons were performed by the Student’s test or one-way ANOVA (ANalysis Of VAriance) using GraphPad Prism 6 (San Diego, CA, USA).

## 3. Results

### 3.1. Double Nickase Cas9 PLAUR Gene Knockout

We transfected melanoma and colon cancer cells with two CRISPR/Cas9 D10A plasmids, each one bearing a specific sgRNA, as shown in Figure 1A, designed by the manufacturer to generate a double strand break in uPAR exon 3 (Figure 1B), thus obtaining a complete *PLAUR* gene knockout. The use of two sgRNA and the mutant version of the Cas9 enzyme will lead to the reduction of unwanted off-target effects, albeit reducing the efficiency as well [24]. We selected uPAR KO cells and exploited the positivity for the GFP marker by Fluorescence-Activated Cell Sorting (FACS) and culturing them with puromycin for 2–3 weeks. The pools of KO cells were diluted limitingly to obtain single clones that were subsequently evaluated for uPAR mRNA expression by qPCR, selecting only the clones with an expression under 0.15-fold of *PLAUR* (Appendix A). Individual clones were then screened by WB for uPAR expression, and from this selection, we obtained one uPAR KO clone from A375p, called hereafter A375 PL1, and one from A375M6 called M6 A5. A375p and A375M6 Control were transfected instead with a plasmid containing a scramble sgRNA. As further internal control, and to avoid tissue specific effects of uPAR deprivation, we decided to also introduce another uPAR KO clone obtained, as described above, from a completely different tissue, the colon carcinoma HCT116 cell line, referred to from now on as HCT116 A3. We evaluated the success of transfection with RT-PCR and WB (Figure 2A,B). We immediately noticed deep morphological changes, as uPAR KO clones showed larger dimension and different shapes, with respect to the cells transfected with the Control Plasmid (Figure 2C). Analyzing the cells dimension, we observed that while A375 PL1 and M6 A5 showed a larger dimension, HCT116 A3 did not increase its average length. However, when also evaluating the cellular complexity by FACS analysis, we evidenced a higher internal complexity in all uPAR KO clones (Appendix A).

### 3.2. uPAR Loss Decreased Cells Glycolytic Capacity

We decided to investigate whether the complete uPAR loss may have triggered a metabolic profile alteration by performing a metabolic stress assay by exploiting the Seahorse platform. We subjected Control and uPAR KO cells to a glycolytic stress test, adding into the cell medium three sequential different treatments (Glucose, Oligomycin and 2-DG) and measuring the variations of the mpH media (expressed as Extra Cellular Acidification Rate—ECAR). After three initial measures and recording the Non-Glycolytic Acidification (NGA), we injected 10 mM Glucose observing an increased variation of the mpH attributable to glycolysis. We then added 1 μM oligomycin in order to completely stop the mitochondrial activity, inhibiting the complex V (ATPase), to record another mpH increase that is referenced as the glycolytic capacity, i.e., the maximum cell ability to perform glycolysis in absence of the mitochondrial activity. Finally, 50 mM of 2-Deoxy-D-glucose (2-DG) was added to completely stop the glycolytic process. Indeed, having had the 2-DG the 2-hydroxyl group replaced by hydrogen, the phosphoglucoisomerase was incapable of completing the reaction, thus observing a decrease in the mpH. The difference between the glycolytic capacity and the glycolysis is commonly referred as the glycolytic reserve. We observed a significant decrease of glycolysis and glycolytic capacity of all the three KO clones (Figure 3), as expected from our previous experiment using anti-uPAR siRNA [25]. To further confirm our results, we reintroduced uPAR expression in the KO cells (Appendix A) using an Okayama-Berg vector containing uPAR cDNA [23], demonstrating that uPAR rescue is sufficient enough to almost restore the glycolytic capability to the control levels. The cell lines with the rescued expression of uPAR are indicated from now with the wording + uPAR.

### 3.3. The Lactate Secretion Paradox

Since it had been reported that several cancer cells benefit from the so-called “Warburg effect” that increases the glycolysis pathway, after studying the glycolytic efficiency, we decided to focus on the evaluation of the uptake of the two most important metabolites, glucose, and its derivative, lactate. The disrupted glycolysis in uPAR KO clones reported above is not sustained by any impairment in glucose uptake mechanisms, since no variations have been observed between Control and uPAR KO cells during the in vitro FITC-NBDG uptake test at flow cytometer (Figure 4A). Despite what we reported about the decreased glycolysis in uPAR KO cancer cells detected via the Seahorse platform, we observed that they release a higher amount, compared to control cells, of lactate—the very final product, together with protons, of the glycolytic pathway—in the extracellular media, as shown in Figure 4B. To explain such a discrepancy, we evaluated the expression levels of Lactate Dehydrogenase A and B (LDHA and LDHB), which are responsible for the conversion of pyruvate to lactate and vice versa, respectively. As shown in Figure 4C, even though we reported a general downregulation, the ratio between the two cDNAs was in favor of *LDHA* in uPAR KO cells, confirming that most of the pyruvate was converted to lactate and less was recycled back. We verified this trend in Western Blot, as shown in Figure 4D, demonstrating that LDHA was upregulated in A375 PL1 while LDHB was downregulated, explaining in such a way the consistent amount of lactate secreted in the medium. Such a difference was reduced in the other two KO clones, where LDHA and LDHB were both downregulated.

### 3.4. uPAR KO Leads to Deep Changes in the Expression Profile of Metabolism-Related Genes

To better understand how uPAR KO may so deeply affect the melanoma and colon carcinoma cells metabolism, we decided to evaluate the glutamine, lactate and glucose transporters. Indeed, glutamine is reported to be a fundamental metabolite, fueling the pool of several intermediate metabolites. For this reason, we decided to check the expression levels of the glutaminases 1 and 2 enzyme as well. As shown in Figure 5, after uPAR KO, there was a downregulation in A375p and the HCT116 clones of the lactate importer gene *MCT-1*, which is one of the main genes responsible for the modulation of the cellular levels of lactate and pyruvate, while the expression level of the lactate exporter, *MCT-4*, was downregulated only for A375, although we did not observe any changes in the other two cell lines. GLUT1 (*SLC2A1*) and GLUT3 (*SLC2A3*) expression levels, as well as *GLS1*, were discordant among the clones, while we reported the higher Glutaminases 2 (*GLS2*) expression in all uPAR KO cells that might fuel the intracellular pool of the Krebs cycle intermediate. Last, we observed a downregulation of ASCT2 (*SLC1A5*) expression in A375 and HCT116 KO clones, while in M6 A5, the variation was not statistically significant. Playing uPAR, an important role for several cellular functions triggering the PI3K/Akt pathway through many membrane receptor partners, we decided to analyze the phosphorylation level of Akt and ERK. We verified that in the absence of uPAR, Akt was significantly less phosphorylated while ERK, to compensate such effect, was strongly phosphorylated. Such changes in the phosphorylation pattern may reflect the deep metabolic alterations observed here.

### 3.5. Spare Respiratory Capacity Enhanced in uPAR KO Cells

To further improve our knowledge of the relationship between uPAR and cell metabolism, we performed a Mitostress assay, measuring, throughout the Seahorse platform, the Oxygen Consumption Rate (OCR) of control cells, uPAR KO cells and the cells obtained with the rescue experiment, and subjecting them to three different treatments (Oligomycin, FCCP and Rotenone/Antimycin A). As for the above described Glycostress test, after three initial measures of valuating the Basal cells respiration, we added 1 μM oligomycin to block oxidative phosphorylation and, thus, obtained the ATP production and the so-called proton leak. Then, 1 μM FCCP, a mitochondrial membrane uncoupler, was added to measure the Spare Respiratory Capacity, i.e., the oxygen reserve that a cell is able to use to respond to a higher energy demand under stress conditions. Lastly, 0.5 μM rotenone/antimycin A was used to completely block mitochondrial oxidative phosphorylation through the inhibition of complex I and III activities. As observed in Figure 6, we reported a general enhancement of the Spare Respiratory Capacity in uPAR KO clones compared to wild-type cells. This capacity reflects the mitochondrial membrane potential, which is normally unused in physiological conditions.

### 3.6. uPAR Loss Impaired Mitochondria Biogenesis

Relying on the OXidative PHOSphorylation (OXPHOS) machinery in the mitochondrial compartment and given the result of the significant upregulation of the SRC in uPAR KO cells, we then examined the control, the uPAR KO and the KO rescued uPAR expression cells ultrastructurally to seek possible mitochondrial alterations (Figure 7A). Despite the noted functional differences, both A375p and A375M6 melanoma cells showed basically similar mitochondria in the different experimental conditions. Only in A375p PL1 cells were elongated mitochondria and autophagic vacuoles with mitochondrial remnants encountered, a feature suggesting increased mitochondrial turnover. We also detected an increased number of mitochondria per cell in the two melanoma cell lines (Figure 7B). On the other hand, colon cancer HCT 116 cells displayed substantial differences; in fact, control cells showed rod-shaped mitochondria with numerous cristae, typical of mature cells, while the HCT116 A3 cells mostly displayed large vesicular mitochondria with a cleared matrix and a reduction of cristae, typical of immature of dysfunctional cells. Of note, these mitochondrial alterations were not detected in the KO cells subjected to rescued uPAR expression.

## 4. Discussion

Here we have exploited the CRISPR/Cas9 gene knockout approach to investigate the relationship between uPAR and the cell metabolism in human melanoma cells as consequences of its total and irreversible loss. We decided to use an alternative version of the common spCas9, called D10A, which is incapable of performing a double strand break due to a mutation in the RuvC domain. This “Nickase,” requiring two sgRNAs to complete the gene knockout, is much more specific than wild-type Cas9 and less prone to off-target effects [26]. However, the cost for such specificity is usually a reduced efficiency leading to the selection of a few number of KO clones. As reported by Rysenkova et al. [27], the proportion of uPAR-positive cells was described to be 64.6%, 59.1% and 46.0%, respectively, after one, two and three rounds of transfection using a canonical Wild-Type Cas9, demonstrating *PLAUR* to be a hard target to be knocked-out. Moreover, A375 and M6 are described by ATCC as hypotriploid cell lines so it is hard for the Cas9 to selectively knockout all the genomic copies of the *PLAUR* gene. Thus, to avoid every possible bias due to tissue specificity and to verify our results, as reported in other CRISPR-based studies [28], we decided to also perform uPAR KO on a colon cancer cell line. We exploited two melanoma cell lines, one primary and the other one metastatic, which were genetically identical but expressed different uPAR levels, to better understand how uPAR expression might be important for cancer cell metabolism [29]. The main observations of our work indicate that *PLAUR* gene KO causes a decreased glycolytic capacity paired with an increased lactate secretion. This paradox is partially explained by analyzing the ratio between LDHA and LDHB. Indeed, even if uPAR KO cells produce less lactate, they are less prone to convert it back in pyruvate. The discrepancy in the lactate levels will deserve further investigations as the biological process that produces lactate might be fueled by several metabolic pathways. However, we are confident that the different methods by which we modulated uPAR expression, here and in our previous works [13,14,15,16], might be responsible for the several differences observed. Indeed, using a siRNA, we triggered an acute effect [25], downregulating uPAR expression for a maximum of 72 h while by exploiting the CRISPR/Cas9, we obtained stable KO cell lines, forcing the cells to adapt to uPAR deprivation. Moreover, we do believe that by losing uPAR mRNA expression, all the miRNAs that are attached to its 3′ UTR, as reported by Li Santi et al. [30], might modulate several important biological functions. Moreover, even though the receptor pattern related to metabolism is not completely coherent among the three cell lines, we need to point out that the most similar expression levels among A375 and HCT116 clones may be due to their nature as primitive tumor-derived cells, while M6 is a cell line obtained by a metastasis in the lungs. The most interesting fact that we want to point out is the GLS behavior. Indeed, while uPAR KO downregulated *GLS1* only in the primitive cell lines (i.e., A375p and HCT116 uPAR KO), *GLS2* was strongly upregulated in all three cell lines. Higher Glutaminases 2 (GLS2) expression in uPAR KO cells might be responsible for the production contribution of Krebs cycle intermediates. As reported by Bolzoni M et al., some types of human tumors exhibit a high requirement for glutamine (“glutamine addiction”) and use large amounts of amino acid as an anaplerotic substrate [31]. Moreover, as reported by Liu et al., GLS2 negatively regulates the PI3K/AKT signaling pathway [32]. It has been widely known that, when uPAR is expressed at low levels, pAkt is downregulated [33] while pERK is strongly upregulated; uPAR loss, in fact, inhibits the PI3K/Akt pathway and forces the cells to use ERK as an alternative pathway to grow and survive [34]. Here, we verified this mechanism reporting that after uPAR loss, the phosphorylation of Akt was inhibited and, to compensate for this effect to keep the cell proliferating and surviving, the phosphorylation levels of ERK were enhanced. Indeed, Akt is reported to play a major role in stimulating aerobic glycolysis in cancer cells [35]. Moreover, we recently observed that uPAR drives a glycolytic phenotype in melanoma cells [25], triggering a Warburg phenotype mediated by the complex α5β1-integrin-uPAR-EGFR. By using siRNA targeting uPAR, we also previously demonstrated a downregulation of PDK1, which normally promotes the phosphorylation of Akt [36,37]. The inhibition of PDK1 not only inhibits the glycolytic profile of cancer cells, but its downregulation may also be responsible for the enhancement of the spare respiratory capacity observed in uPAR KO cancer cells. Indeed, it was reported that the Pdks protein family is a negative regulator of the spare respiratory capacity [38]. Thus, with our work, we generated a stable in vitro model of uPAR deprivation in cancer cells that allowed us to obtain important information on the two main energy production pathways, i.e., glycolysis and OXPHOS. Being a uPAR receptor, which is commonly modulated not only by hypoxia, cytokines and transcription factors such as NF-kB and TCF/LEF [39] but also by cell-cell contact [40], it is of fundamental importance to study cancer cell metabolism with cells always expressing the same amount of uPAR. Lastly, all the above described metabolic changes were reflected in an impaired mitochondrial biogenesis. Indeed, while the two melanoma cell lines did not show evident differences in mitochondria ultrastructural morphology, we observed a higher number of mitochondria in A375 PL1 and M6 A5, maybe reflecting an increased mitochondrial turnover. Conversely, HCT116 A3 showed the typical features of immature mitochondria, including large vesicular mitochondria with a cleared matrix and reduced cristae [41].

## 5. Conclusions

In our studies, we confirmed the relationship between uPAR and metabolism, which has already been partially observed by Gao C et al. [42], using FACS sorted cells for uPAR from a small cell lung cancer cell line. We validated our previous work proving that uPAR overexpression plays a central role in the Warburg-like metabolism of cancer cells.

## Figures and Tables

**Figure 1 cells-09-00308-f001:**
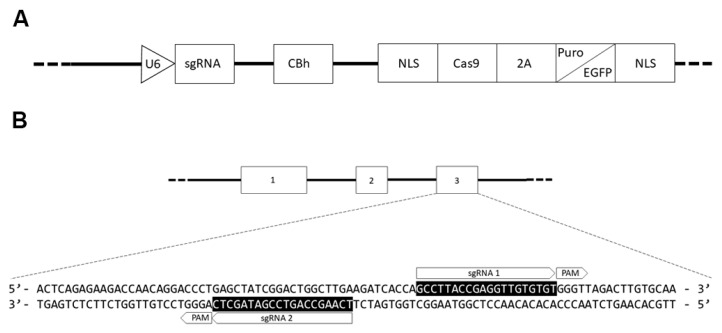
(**A**) The two plasmids have the same structure except for the sgRNAs, which are designed to be complementary to the exon 3 of *PLAUR* gene (**B**), and the markers bearing Puromycin resistance and the Enhanced-GFP. Such plasmids were tested and verified by the manufacturer.

**Figure 2 cells-09-00308-f002:**
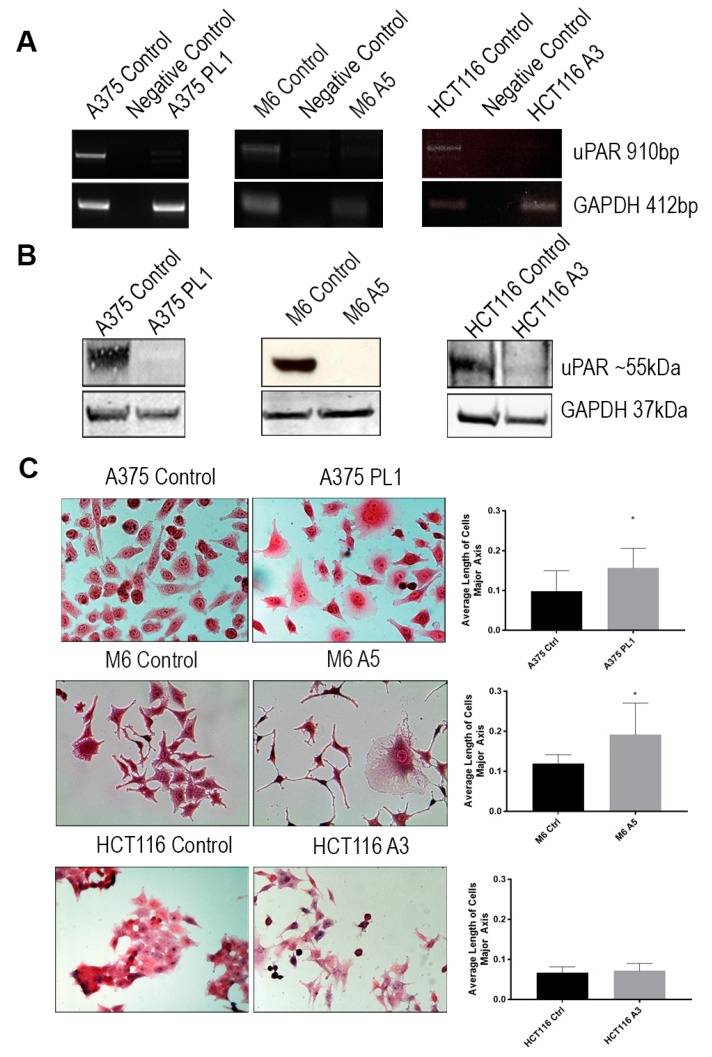
(**A**) Total RNA isolated was subjected to Reverse Transcriptase-PCR analysis of *PLAUR* expression, and *GAPDH* was used as a loading control (*n* = 3). (**B**) Whole cell lysates were analyzed by Western Blot for uPAR expression, and GAPDH was used as a loading control (*n* = 3). (**C**) Images of Control and uPAR KO cells 2 weeks after transfection. Cells were fixed and stained with Hematoxylin and Eosin. Images were captured at 10× magnification and the cells major axis was analyzed by ImageJ (*n* = 15) Data are presented as mean ± SD. * *p* < 0.01 (Student’s test).

**Figure 3 cells-09-00308-f003:**
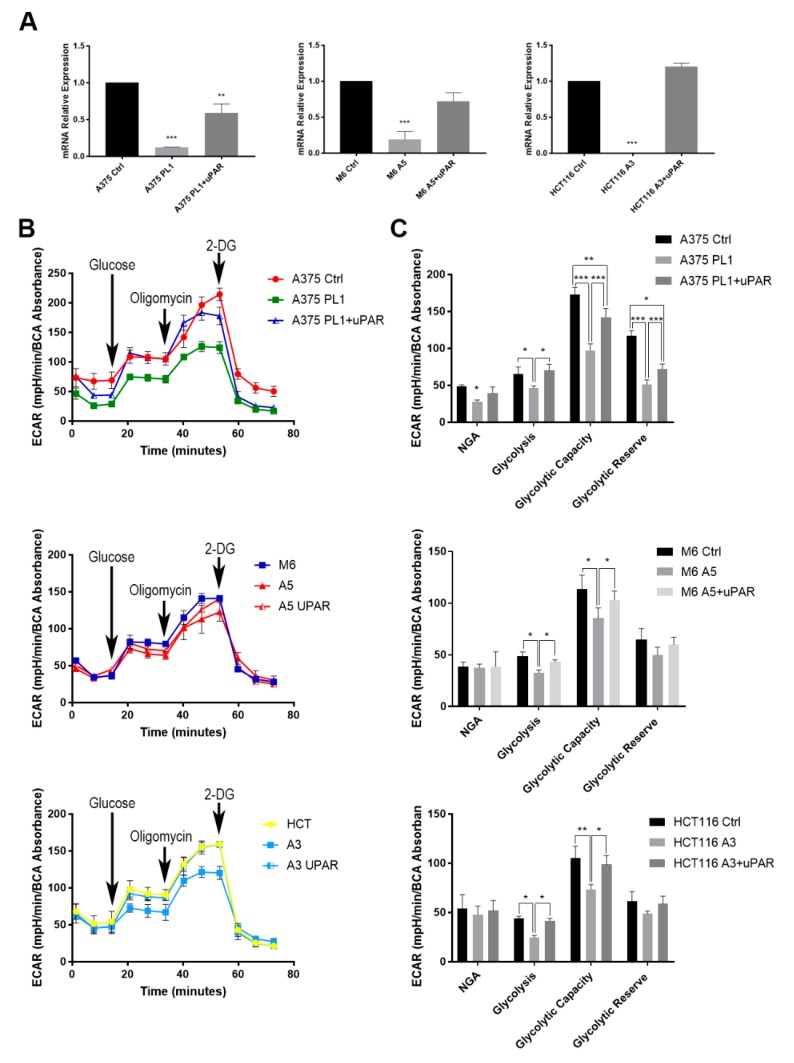
(**A**) Total RNA isolated was subjected to quantitative PCR (qPCR) analysis of *PLAUR* expression. Data are presented as mean ± SD. ** *p* < 0.01; *** *p* < 0.001 (One-way ANOVA test) (*n* = 3). (**B**) The Glycostress assay was performed as described in “Results.” The ExtraCellular Acidification Rate (ECAR) (**C**) measured in Seahorse Glycostress assays is shown as traces from a representative Seahorse run. (**B**) The average of the three measures after the exposure to the above described drugs is summarized in histograms. Here are the summarized results from at least three independent biological experiments, each consisting of independent cell plating on five Seahorse microplate wells. Data are presented as mean ± SD. * *p* < 0.05; ** *p* < 0.01; *** *p* < 0.001 relative to Control (One-way ANOVA test).

**Figure 4 cells-09-00308-f004:**
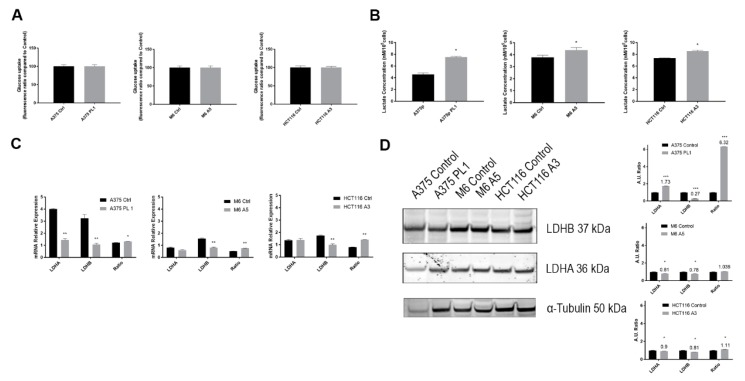
(**A**) Quantification of the fluorescence probe (2-NBDG) uptake through flow cytometer analysis (*n* = 3). (**B**) Lactate production in conditioned media collected after 24 h (*n* = 5). (**C**) Total RNA isolated was subjected to qPCR analysis of *LDHA* and *LDHB* expression. The ratio represents the LDHA/LDHB expression (*n* = 3). (**D**) Immunoblot analysis of total cell lysates in order to determine the expression levels of LDHA and LDHB. Expression levels were checked by loading equal amounts of total protein. Densitometry analysis was performed using ImageJ software (*n* = 3). Data are presented as mean ± SD. * *p* < 0.05; ** *p* < 0.01; *** *p* < 0.001 (Student’s test).

**Figure 5 cells-09-00308-f005:**
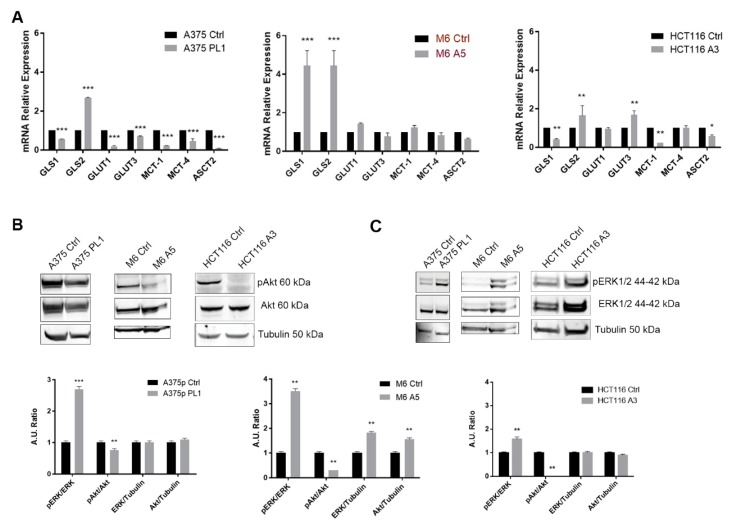
(**A**) Total RNA isolated was subjected to qPCR analysis of several of the main metabolism-related genes (*n* = 3). (**B**) Immunoblot analysis of total cell lysates to determine the phosphorylation and the expression of Akt and (**C**) ERK (*n* = 3). Expression levels were checked by loading equal amounts of total protein. Densitometry analysis was performed using ImageJ software. Data are presented as mean ± SD. * *p* < 0.05; ** *p* < 0.01; *** *p* < 0.001 (Student’s test).

**Figure 6 cells-09-00308-f006:**
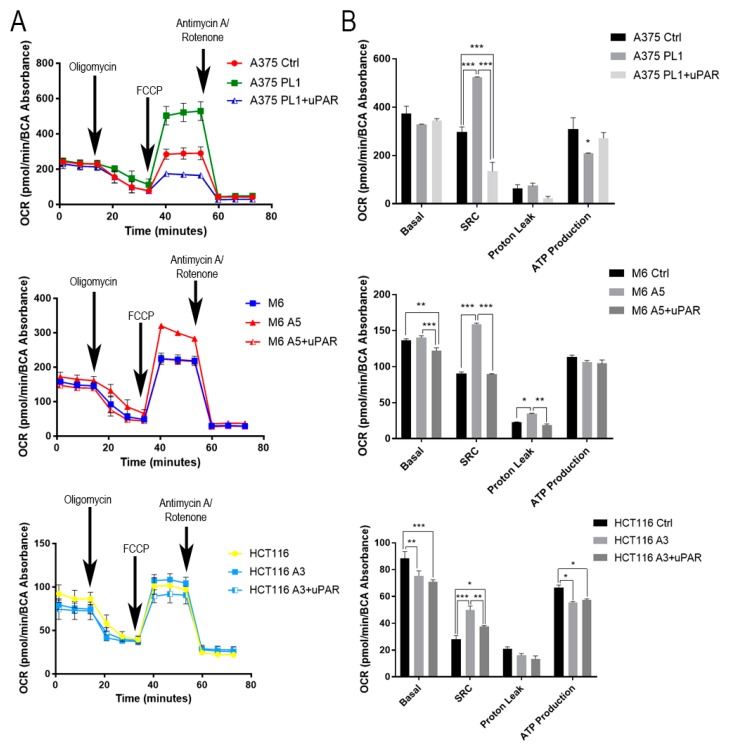
The Mitostress assay was performed as described in “Results.” Oxygen Consumption Rate (OCR) (**A**) measured in Seahorse Mitostress assays is shown as traces from a representative Seahorse run. (**B**) The average of the three measures after the exposure to the above described drugs is summarized in histograms. Here are the summarized results from at least three independent biological experiments, each consisting of independent cell plating on five Seahorse microplate wells. Data are presented as mean ± SD. * *p* < 0.05; ** *p* < 0.01; *** *p* < 0.001 relative to Control (One-way ANOVA test).

**Figure 7 cells-09-00308-f007:**
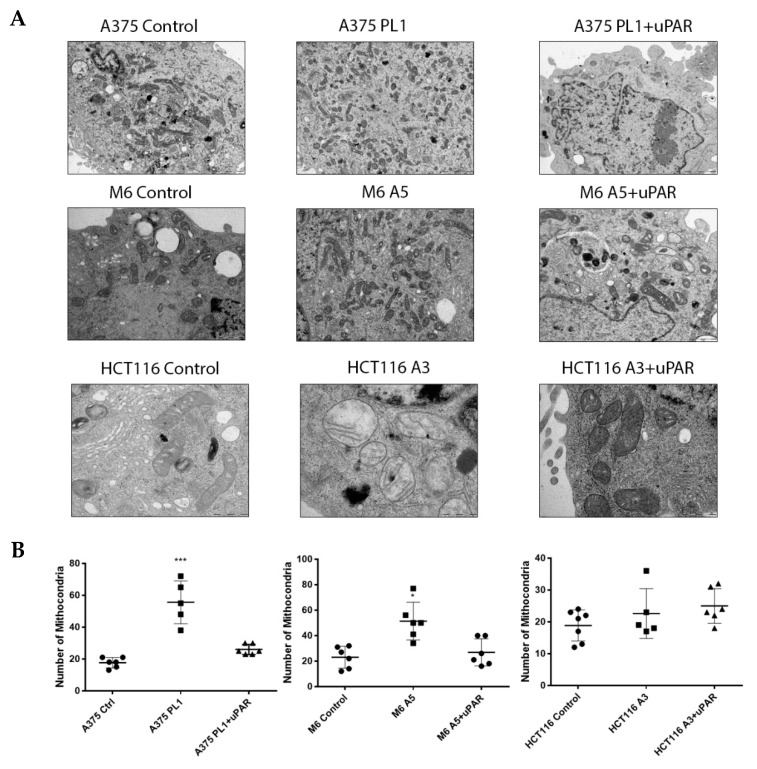
(**A**) Ultrastructural analysis of Control, uPAR KO and uPAR “rescued” cells using transmission electron microscopy. Representative images are shown at magnification of ×12,000 for A375p and M6, and ×50,000 for HCT116. (**B**) Number of mitochondria per cell. Data are presented as mean ± SD. * *p* < 0.05; ** *p* < 0.01; *** *p* < 0.001 relative to Control (One-way ANOVA test).

**Table 1 cells-09-00308-t001:** Off-target sites evaluation.

Bulge Type	Target	Chromosome	Position	Direction	Mismatches	Bulge Size
**X**	crRNA: TCAAGCCAGTCCGATAGCTCNGG DNA: TCAAGCCAGTCCGATAGCTCAGG	chr19	43665375	+	0	0
**X**	crRNA: GCCTTACCGAGGTTGTGTGTNGG DNA: GCCTTACCGAGGTTGTGTGTGGG	chr19	43665343	−	0	0

**Table 2 cells-09-00308-t002:** List of all primers used.

Gene	Sense	Antisense
*GAPDH*	CAATGACCCCTTCATTGACCTC	AGCATCGCCCCACTTGATT
*β2-M*	GCCGTGTGAACCATGTGACT	GCTTACATGTCTCGATCCCACTT
*uPAR*	GGTCACCCGCCGCTG	CCACTGCGGGTACTGGACA
*GLS1*	TGCTACCTGTCTCCATGGCT	CCTAGATGGCACCTCCTTT
*GLS2*	TGCCTATAGTGGCGATGTCTCA	GTTCCATATCCATGGCTGACAA
*GLUT1*	CGGGCCAAGAGTGTGCTAA	TGACGATACCGGAGCCAATG
*GLUT3*	CGAACTTCCTAGTCGGATTG	AGGAGGCACGACTTAGACAT
*MCT-1*	GTGGCTCAGCTCCGTATTGT	GAGCCGACCTAAAAGTGGTG
*MCT-4*	CAGTTCGAGGTGCTCATGG	ATGTAGAGGTGGGTCGCATC
*ASCT2*	GGTGGCTGGCAAGATCGT	CCAAGGCGGGCAAAGAG
*LDHA*	AGCCCGATTCCGTTACCT	CACCAGCAACATTCATTCCA
*LDHB*	CTAGATTTCGCTACCTTAT	TCATTGTCAGTTCCCATT
*MGSO*	TGCACCATCTGTCACTCTGTTAACCTC	
*GPO1*	ACTCCTACGGGAGGCAGCAGTA

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
