# Peer review of "uPAR Knockout Results in a Deep Glycolytic and OXPHOS Reprogramming in Melanoma and Colon Carcinoma Cell Lines"

_cells, 2020, doi:10.3390/cells9020308_

Round 1

Reviewer 1 Report

In the manuscript by Biagioni et al. entitled “uPAR Knockout Results in a Deep Glycolytic and OXPHOS Reprogramming in Melanoma and Colon Carcinoma”, the authors investigated the role of uPAR knockout established by CRISPR/Cas9 on the metabolism of human melanoma and colon cancer cell lines. The manuscript provides some interesting aspects about uPAR role in cancer. The work deserves to be considered for potential publication; however, the manuscript should be thoroughly re-written and some important concerns should be raised:

Major remarks:

Why did you select for the consequent study only a single clone from each cell type (A375PL1 and M6A5)? Do A375p and A375M6 genetically differ one from another? As far as I concerned, A375M6 posses invasive phenotype so what was the goal to select these cells for the study? Did authors perform a Sanger sequencing to prove uPAR KO in their selected clones? Please also clarify the methods used to confirm the absence of off-target activity of the CRISPR/Cas9 system : “No off-target sites in the loci of the most likely off-target activity of the CRISPR/Cas9 system targeted by the chosen sgRNAs could be detected” and provide the results. Does the basal level of uPAR expression differ in selected cell lines? The idea of the paper is difficult to follow. Please, describe the evolution of your hypothesis/ideas in each Results section, rather then directly pass to the obtained results without any explanations. Although it is stated that “Multiple comparisons were performed by the Student test

or One-way or Two-way ANOVA” in Materials and Methods section, all the figures are labelled “two-way ANOVA”. Please explain, which factors where tested each time by 2-way ANOVA. For now, it seems that 2-way ANOVA test was incorrectly performed on Figures 3-6 for two-group or multiple-group comparisons with only one categorical variable (factor). If a cell-line was used as categorical factor, then please provide the appropriate plots with 2 categorical factors accordingly to analysis that was performed.

Could authors provide statistic data of  the cell morphology changes (after uPAR-KO) since most of the cancer cell types vary in terms of cell shape even within total cell population (Fig 2 C)? Did the authors perform a statistical analysis to calculate the differences between uPAR KO and +uPAR (figure 3)? Please add more details to confirm the conclusion that “uPAR rescue is sufficient to restore the glycolytic capability almost to the control levels.” LDHA and LDHB should be measured at protein level (by Western blot, for example), to prove that their ratio is upregulated in uPAR KO cells. I would suggest to measure cellular pyruvate concentration and calculate cellular lactate/pyruvate ratios to provide more details on “lactate secretion paradox” and confirm that in uPAR KO most of the pyruvate is converted to lactate and not recycled back. Please comment why GAPDH was used as a loading control since it mRNA could also be changed as glycolysis-associated gene particularly in cancer cells. The authors need to explore the mechanism through which uPAR affects cellular metabolism. Whether the effects they observe depend on uPA? For example, the authors could investigate whether inhibition of uPAR binding to its primary ligand uPA (using blocking antibodies or inhibitors) is similar to uPAR KO? Whether the addition of soluble recombinant uPAR to uPAR-KO cells would restore glycolysis and oxphos? It is known that uPAR can induce cellular signaling through the association with transmembrane receptors. What is the key cellular pathway affected by uPAR KO that induces the observed decrease in cellular metabolism? What is the role of uPAR KO on HIF-1a expression/activity and other important metabolic transducers? Does uPAR KO change glucose dependence or resistance of cancer cells to hypoxic conditions? Please comment on why in the previous paper by the lab uPAR silencing resulted in decreased glucose uptake and decreased lactate production (Laurenzana et al., 2017), which is not confirmed by uPAR KO in the current manuscript. Please comment on why using CRISPR/Cas9 for uPAR KO in some clones led to significant upregulation of uPAR mRNA. How do authors explain so negligible difference of LDHA/LDHB ratio in Fig.4C? Does it have any functional effect?

Minor remarks:

Please proof-read English with a native English-speaking scientist. The names for clones are a bit confusing: PL1, A5, A3 and are used not uniformly throughout the manuscript. Besides, the names differ from the coding used in Supplementary material. As only one clone is selected for each cell line, I would suggest to refer to them as cell line-ΔuPAR, cell line-uPAR KO or similarly and describe which clone was selected in Material and Methods section. All the figures lack explanations of the labelling and abbreviations, e.g. +uPAR – is it uPAR rescue? In each case what differences are indicated by * - relative to control? Figure 3 is difficult for understanding. I would suggest to indicate on the plots with ECAR acquisition at different time points (A) the moments of glucose, oligomycin and 2-DG addition. Zones of metabolic activity referred in the text as NGA, Glycolysis, glycolytic capacity, glycolytic reserve could be also visualized. Please also explain in more detail how the values on histograms B were calculated. The same should be done with Figure 5. I would also suggest to use the same coloring for WT, uPAR KO and uPAR rescue cells in different cell lines as it is difficult to compare between two red and two blue curves. I would suggest to reformulate the phrase about upregulated LDHA/LDHB ratio in uPAR KO cells “confirming that most of the pyruvate is converted to lactate and not recycled back”. LDHA can catalyze both reactions, it just has a higher affinity for pyruvate, preferentially but not ultimately converting pyruvate to lactate. Authors report that Spare Respiratory Capacity is enhanced in uPAR KO cells, however they do not comment on the fact that in two primary cell lines the ATP-production linked respiration is decreased in uPAR KO. Can the authors conclude that uPAR KO results in less metabolically active cells? Many terms are not uniformly used throughout the manuscript, e.g. “The Glycostress assay”, “Glycolysis Stress Assay”, “glycolytic stress test” or “Seahorse”, “SeaHorse” or “Sea Horse”. Please correct that. Please indicate how many replicates were used for each test in Materials and Methods section. The principle primers’ selection in Table1 is not postulated in the Introduction or the Discussion section. There are a lot of typographical errors in the text. Some of them: “104 cell/well”, line 119 and 129  - gRNA and sgRNA, NGA abbreviation in Fig. 3 legend, what is FCCP stand for, line 229 turn-over should be replaced by turnover etc…

Author Response

Thanks for your valuable suggestions, we put every effort to improve our manuscript following your concerns and requests.

Why did you select for the consequent study only a single clone from each cell type (A375PL1 and M6A5)?

We selected only one clone for each cell line due to the double Nickase low efficiency, as reported in the Discussion section. Indeed, we strongly optimized the transfection conditions due to a great difficulty in obtaining uPAR-KO cells. As yet reported by Rysenkova et al. [CRISPR/Cas9 nickase mediated targeting of urokinase receptor gene inhibits neuroblastoma cell proliferation. Oncotarget. 2018 Jun 29;9(50):29414-29430.] the proportion of uPAR-positive cells was described to be 64.6%, 59.1%, and 46.0%, respectively after one, two and three rounds of transfection using a canonical Wild-Type Cas9, demonstrating PLAUR to be a hard target to be knocked-out. Moreover, A375 and M6 are described by ATCC as hypotriploid cell lines so it is tough for the Cas9 to knockout selectively all the genomic copies of the PLAUR gene.

Do A375p and A375M6 genetically differ one from another? As far as I concerned, A375M6 posses invasive phenotype so what was the goal to select these cells for the study?

Being M6 a cell line derived from a metastatic lung node obtained by intravenous inoculation of A375 cells in nude mice, the two cell lines are genetically identical (we recently proved it by STR profile). Albeit such genetic identity, they show a very different behavior in terms of migration, invasion and thus uPAR expression, and this is the main reason why we decided to exploit such cells lines in our study.

Did authors perform a Sanger sequencing to prove uPAR KO in their selected clones?

Given the results reported in Figure 2A and 2B, i.e. the complete loss of uPAR expression, in terms of both mRNA and protein, we did not evaluate as necessary a Sanger sequencing. Moreover, we added a cytometer analysis to the Supplementary Material (Supplementary Figure 2A) to further prove the success of uPAR KO.

Please also clarify the methods used to confirm the absence of off-target activity of the CRISPR/Cas9 system : “No off-target sites in the loci of the most likely off-target activity of the CRISPR/Cas9 system targeted by the chosen sgRNAs could be detected” and provide the results.

We showed the absence of off-target activity of the designed sgRNA exploiting the more up-to-date Cas-OFF Finder software, including the results of the analysis in the Material and Methods section, as requested.

Does the basal level of uPAR expression differ in selected cell lines?

The different uPAR expression level, as a reason for the cell model choice, was added to the Discussion citing the manuscript where they were originally evaluated [Laurenzana A et al. EGFR/uPAR interaction as druggable target to overcome vemurafenib acquired resistance in melanoma cells. EBioMedicine. 2019.], while HCT116 cell line was added to overcome any possible bias due to tissue specific effects.

The idea of the paper is difficult to follow. Please, describe the evolution of your hypothesis/ideas in each Results section, rather then directly pass to the obtained results without any explanations.

To make the paper easier to read, we added in each Result section a brief discussion of our hypothesis, as requested.

Although it is stated that “Multiple comparisons were performed by the Student test or One-way or Two-way ANOVA” in Materials and Methods section, all the figures are labelled “two-way ANOVA”. Please explain, which factors where tested each time by 2-way ANOVA. For now, it seems that 2-way ANOVA test was incorrectly performed on Figures 3-6 for two-group or multiple-group comparisons with only one categorical variable (factor). If a cell-line was used as categorical factor, then please provide the appropriate plots with 2 categorical factors accordingly to analysis that was performed.

Following your precious suggestions, we reviewed again all the experimental data, finding a wrong report of the statistical analysis methods which were thus revised and corrected accordingly.

Could authors provide statistic data of  the cell morphology changes (after uPAR-KO) since most of the cancer cell types vary in terms of cell shape even within total cell population (Fig 2 C)?

We added a graph (2C) reporting the measures of the major axis of the cells, explaining in the text that we observed an increase dimension only in the two melanoma cell lines. Moreover, we added to the Supplementary Figure 2 a cloud dot plot reporting also the internal cell complexity analyzing the cells through flowcytometry.

Did the authors perform a statistical analysis to calculate the differences between uPAR KO and +uPAR (figure 3)?

The statistical analysis of the comparison between uPAR KO and uPAR+ was added as requested.

Please add more details to confirm the conclusion that “uPAR rescue is sufficient to restore the glycolytic capability almost to the control levels.” LDHA and LDHB should be measured at protein level (by Western blot, for example), to prove that their ratio is upregulated in uPAR KO cells.

As demonstrated in Figure 3 and 6, Glycolysis, Glycolytic Capacity as well as the Spare Respiratory Capacity were altered by uPAR KO while the reintroduction of uPAR gene restored such capacities almost to the control levels. Being the Seahorse assay a functional test and being currently the gold standard to evaluate the cellular metabolic capabilities, we did not evaluate as necessary to check also all the glycolysis- and Krebs cycle-related protein levels and the relative intermediate metabolites. Moreover, as yet published in our previous manuscript [Laurenzana A, Chillà A, Luciani C, Peppicelli S, Biagioni A, Bianchini F, Tenedini E, Torre E, Mocali A, Calorini L, Margheri F, Fibbi G, Del Rosso M. uPA/uPAR system activation drives a glycolytic phenotype in melanoma cells. Int J Cancer. 2017 Sep 15;141(6):1190-1200.] we have already demonstrated, with a different approach, that uPAR is mandatory for the glycolytic capacity of cancer cells, triggering the so-called “Warburg effect”.

I would suggest to measure cellular pyruvate concentration and calculate cellular lactate/pyruvate ratios to provide more details on “lactate secretion paradox” and confirm that in uPAR KO most of the pyruvate is converted to lactate and not recycled back.

We did not evaluate as necessary a cellular pyruvate concentration but we provided more data about the lactate secretion paradox adding a Western Blot evaluating the protein level of LDHA and LDHB. However we are very grateful to the referee to its precious suggestion which will be considered in future studies.

Please comment why GAPDH was used as a loading control since it mRNA could also be changed as glycolysis-associated gene particularly in cancer cells.

GAPDH was used as loading control only for the Western Blot in Figure 2 since for the qPCRs was used the β2-Microglobulin as housekeeper gene as reported in Materials and Methods. We used the GAPDH in the first Western Blot due to the molecular weight of uPAR (55 kDa) which may interfere with the more classical α-Tubulin (50 kDa). Moreover, the uPAR monoclonal antibody can be used only in native conditions without the reduction of the protein so the lack of this step reduced the number of housekeeper marker usable in Western Blot. We verified also the reliability of the demonstrated KO clones adding the FACS analysis of CD87 (uPAR) in the Supplementary Figure 2.

We realized that in some legends was erroneously reported the GAPDH as loading control and thus it was corrected. However, we verified at the beginning of the study that GAPDH, β2-Microglobulin and 18S are equally reliable as housekeeper genes since we did not observe any significant variation in their expression pre- and post-uPAR KO.

The authors need to explore the mechanism through which uPAR affects cellular metabolism. Whether the effects they observe depend on uPA? For example, the authors could investigate whether inhibition of uPAR binding to its primary ligand uPA (using blocking antibodies or inhibitors) is similar to uPAR KO? Whether the addition of soluble recombinant uPAR to uPAR-KO cells would restore glycolysis and oxphos? It is known that uPAR can induce cellular signaling through the association with transmembrane receptors. What is the key cellular pathway affected by uPAR KO that induces the observed decrease in cellular metabolism? What is the role of uPAR KO on HIF-1a expression/activity and other important metabolic transducers? Does uPAR KO change glucose dependence or resistance of cancer cells to hypoxic conditions

About the relationship between uPAR, uPA and Hypoxia in cancer cell metabolism, it has been yet elucidated in our previous paper [Laurenzana A, Chillà A, Luciani C, Peppicelli S, Biagioni A, Bianchini F, Tenedini E, Torre E, Mocali A, Calorini L, Margheri F, Fibbi G, Del Rosso M. uPA/uPAR system activation drives a glycolytic phenotype in melanoma cells. Int J Cancer. 2017 Sep 15;141(6):1190-1200.], demonstrating that the binding of uPA to uPAR increased PKM2 expression, but not affecting PKM1, the lactate production in M6 cells was enhanced by uPAR activation with uPA while the stimulation with uPA in siPLAUR cells did not produce any increment in lactate extrusion. Further, MCT4 expression was found to be under the control of uPAR and such effect was amplified upon uPA stimulation. HIF1α transactivates genes whose products mediate tumor invasion and glycolytic metabolism providing the common denominator between tumor metabolism and the phenotypic invasive features.

We verified our hypothesis upon ERK and Akt phosphorylation levels, as suggested and such results were added in Figure 5 and in the relative text sections.

Please comment on why in the previous paper by the lab uPAR silencing resulted in decreased glucose uptake and decreased lactate production (Laurenzana et al., 2017), which is not confirmed by uPAR KO in the current manuscript.

Regarding the discrepancy in the lactate levels we know we will need to further investigate such phenomenon; however, we are confident that the different methods by which we modulated uPAR expression, here and in our previous work, might be responsible for the several differences observed. Indeed, using a siRNA we obtained an acute effect, downregulating uPAR expression for a maximum of 72h while exploiting the CRISPR/Cas9 we obtained a stable KO cell lines, forcing the cells to adapt to uPAR deprivation. Moreover, we are confident that loosing uPAR mRNA, all the miRNAs that are attached to its 3’ UTR, as reported by Li Santi et al. [Li Santi A, Gorrasi A, Alfieri M, Montuori N, Ragno P. A novel oncogenic role for urokinase receptor in leukemia cells: molecular sponge for oncosuppressor microRNAs. Oncotarget. 2018 Jun 12;9(45):27823-27834], might modulate several biological functions which will be the topic of our future investigations.

Please comment on why using CRISPR/Cas9 for uPAR KO in some clones led to significant upregulation of uPAR mRNA.

As the referee suggested, some uPAR KO clones during the preliminary screening showed upregulated levels of uPAR: limiting diluting clones from a pool of transfected cells may results that some of them show an upregulated expression levels compared to the control, being the tumor cell population composed by a heterogeneous pool of cells. Indeed, for such reason, we screened several single cell-derived clones by qPCR and the clones with an expression under 0.15 were screened again by WB and qualitative PCR analyzing the mRNA in their entire length.

How do authors explain so negligible difference of LDHA/LDHB ratio in Fig.4C? Does it have any functional effect?

As requested, we checked the protein level of LDHA and LDHB verifying that this phenomenon may be responsible for the great difference in lactate secretion for A375 PL1 while in the other two cell lines, even if reduced, we observed the downregulation of both enzymes still in favor of the LDHA at least for the HCT116 A3.

Minor remarks:

Please proof-read English with a native English-speaking scientist.

The manuscript, as requested, was entirely revised

The names for clones are a bit confusing: PL1, A5, A3 and are used not uniformly throughout the manuscript. Besides, the names differ from the coding used in Supplementary material. As only one clone is selected for each cell line, I would suggest to refer to them as cell line-ΔuPAR, cell line-uPAR KO or similarly and describe which clone was selected in Material and Methods section.

The use of the name of the clones was revised in the manuscript in order to be more uniformly used and the relative Supplementary Material legend was better explained.

All the figures lack explanations of the labelling and abbreviations, e.g. +uPAR – is it uPAR rescue? In each case what differences are indicated by * - relative to control?

The coding for the abbreviation was added in the text. The statistical analysis was improved following your suggestions.

Figure 3 is difficult for understanding. I would suggest to indicate on the plots with ECAR acquisition at different time points (A) the moments of glucose, oligomycin and 2-DG addition. Zones of metabolic activity referred in the text as NGA, Glycolysis, glycolytic capacity, glycolytic reserve could be also visualized. Please also explain in more detail how the values on histograms B were calculated. The same should be done with Figure 5.

Figure 3 and 6 were modified including the drug addiction times. Moreover, as reported in the Results the histogram in Figures 3C and 6B were calculated as the average of the three measures obtained by the Seahorse platform after the addition of the selected drugs normalizing the results on the protein content.

I would also suggest to use the same coloring for WT, uPAR KO and uPAR rescue cells in different cell lines as it is difficult to compare between two red and two blue curves.

We did not evaluate necessary to change the graph colors as the same data are also expressed with histograms.

I would suggest to reformulate the phrase about upregulated LDHA/LDHB ratio in uPAR KO cells “confirming that most of the pyruvate is converted to lactate and not recycled back”. LDHA can catalyze both reactions, it just has a higher affinity for pyruvate, preferentially but not ultimately converting pyruvate to lactate.

Such phrase was reformulated following your suggestion.

Authors report that Spare Respiratory Capacity is enhanced in uPAR KO cells, however they do not comment on the fact that in two primary cell lines the ATP-production linked respiration is decreased in uPAR KO. Can the authors conclude that uPAR KO results in less metabolically active cells?

We agree with the referee that uPAR KO cells may be less metabolically active and for such reason we speculate that such mechanism might be one of the responsible for the minimal residual disease. Indeed, in vivo, cancer cells may downregulate uPAR, decreasing their glycolytic capacity, decreasing the proliferation ratio but becoming more resistant to environmental stresses upregulating their SRC.

Many terms are not uniformly used throughout the manuscript, e.g. “The Glycostress assay”, “Glycolysis Stress Assay”, “glycolytic stress test” or “Seahorse”, “SeaHorse” or “Sea Horse”. Please correct that.

All the terms were checked and corrected to be uniformly used.

Please indicate how many replicates were used for each test in Materials and Methods section.

The number of replicates were added in the relative legends or otherwise expressed in the Materials and Methods section.

The principle primers’ selection in Table1 is not postulated in the Introduction or the Discussion section.

The principle of selection was added in the relative paragraph of Materials and Methods.

There are a lot of typographical errors in the text. Some of them: “104 cell/well”, line 119 and 129  - gRNA and sgRNA, NGA abbreviation in Fig. 3 legend, what is FCCP stand for, line 229 turn-over should be replaced by turnover etc…

All the evidenced mistakes were corrected.

Reviewer 2 Report

Biagioni and colleagues investigated the effects of uPAR Knockout on glycolytic and oxidative metabolism in melanoma and colon carcinoma cell lines. They report an increased number of mitochondria in two A375 melanoma cell lines with uPAR knockout, while biogenesis of mitochondria in a HCT116 colon carcinoma culture is impaired (or at least there are morphological differences, which might suggest impaired biogenesis). Mitochondrial spare respiratory capacity was enhanced, and glycolysis decreased with increased secretion of lactate by all uPAR KO cells. They speculate that this discrepancy might be explained by an impaired ratio between LDHA and LDHB.

Overall the manuscript is well written, the data are convincing. The rescue is supporting specificity of the observations. Some overstatements should be removed e.g.:

The title is misleading as the investigations are made in cell lines not melanoma and colon cancer.

Mitochondrial abnormal morphology does not mean impaired biogenesis. In fact, some of the HCT116 cells in Figure 2 look very much apoptotic.

Given the small differences in the LDHA/LDHB ratios, it seems unlikely to cause all the observed differences.

Minor comments:

Line 82: “cDNA sintesys kit” typing error.

Line 93: “104 cells/well” typing error.

Line 114: Students test is not appropriate for multiple comparisons, but according to Figure legends had not been used?

Line 155 and following: “In last” spelling error.

Line 196: “downregulation of ASCT2 expression in all the ko clones” is overstated as it is not significant in in M6 A5.

Figure 5 would be easier to appreciate when the scaling of y-axes would be similar.

The link to the supplementary file is inactive. Anyway, expression levels of the “rescue” experiment should be included in the main file.

Author Response

We are very grateful for your valuable suggestions in order to improve the quality of our manuscript.

The title is misleading as the investigations are made in cell lines not melanoma and colon cancer.

Following your requests, we modified the manuscript title

Mitochondrial abnormal morphology does not mean impaired biogenesis. In fact, some of the HCT116 cells in Figure 2 look very much apoptotic.

Even though the cells morphology in Figure 2 showed some apoptotic-like cells, we previously verified with a Trypan Blue assay that the samples prepared for the TEM analysis contained only vital cells (more than 98% of the cells were vital just before fixing them), thus we are confident that the mitochondrial abnormalities were due only to an impaired biogenesis or to an increased mitochondrial turnover.

Given the small differences in the LDHA/LDHB ratios, it seems unlikely to cause all the observed differences.

Regarding the LDHA/B ratio about the “Lactate Paradox” we assayed the mRNA levels on several biological replicates (N=3) and analyzing their modulation post-KO we evidenced that, even if small, it is statistically significative (p=0.02 for A375, p=0.0015 for M6 and p= 0.002 for HCT116). We checked the protein level too verifying that this phenomenon may be responsible for the great difference in lactate secretion for A375 PL1 while in the other two cell lines, even if reduced, we observed the downregulation of both enzymes still in favor of the LDHA at least for the HCT116 A3.

Minor comments:

Line 82: “cDNA sintesys kit” typing error.

Line 93: “104 cells/well” typing error.

Line 114: Students test is not appropriate for multiple comparisons, but according to Figure legends had not been used?

Line 155 and following: “In last” spelling error.

Line 196: “downregulation of ASCT2 expression in all the ko clones” is overstated as it is not significant in in M6 A5.

We accomplished all the minor remarks modifications including the correction of the typing mistakes and the inappropriate reporting in the Statistical Analysis of the use of the Two-Way ANOVA test.

Figure 5 would be easier to appreciate when the scaling of y-axes would be similar.

y-axis of Figure 5 was modified to be better appreciable.

The link to the supplementary file is inactive. Anyway, expression levels of the “rescue” experiment should be included in the main file.

We added the expression levels of the “rescue experiment” on Figure 3 following your suggestion.

Round 2

Reviewer 1 Report

I would like to see

1. "As yet reported by Rysenkova et al. [CRISPR/Cas9 nickase mediated targeting of urokinase receptor gene inhibits neuroblastoma cell proliferation. Oncotarget. 2018 Jun 29;9(50):29414-29430.] the proportion of uPAR-positive cells was described to be 64.6%, 59.1%, and 46.0%, respectively after one, two and three rounds of transfection using a canonical Wild-Type Cas9, demonstrating PLAUR to be a hard target to be knocked-out. Moreover, A375 and M6 are described by ATCC as hypotriploid cell lines so it is tough for the Cas9 to knockout selectively all the genomic copies of the PLAUR gene."

and

2. " Regarding the discrepancy in the lactate levels we know we will need to further investigate such phenomenon; however, we are confident that the different methods by which we modulated uPAR expression, here and in our previous work, might be responsible for the several differences observed. Indeed, using a siRNA we obtained an acute effect, downregulating uPAR expression for a maximum of 72h while exploiting the CRISPR/Cas9 we obtained a stable KO cell lines, forcing the cells to adapt to uPAR deprivation. Moreover, we are confident that loosing uPAR mRNA, all the miRNAs that are attached to its 3’ UTR, as reported by Li Santi et al. [Li Santi A, Gorrasi A, Alfieri M, Montuori N, Ragno P. A novel oncogenic role for urokinase receptor in leukemia cells: molecular sponge for oncosuppressor microRNAs. Oncotarget. 2018 Jun 12;9(45):27823-27834], might modulate several biological functions which will be the topic of our future investigations."

and

3. discussion about Akt and Erk signaling changes in discussion section.

I believe this will significantly enforce the discussion of the manuscript.

All the other answers of the questions, as well as the experimental work that the authors did, fully satisfied me.

I believe that the article can be published after introducing the information into the discussion section that I proposed.

Author Response

Thanks again for your valuable suggestions.

I would like to see

"As yet reported by Rysenkova et al. [CRISPR/Cas9 nickase mediated targeting of urokinase receptor gene inhibits neuroblastoma cell proliferation. Oncotarget. 2018 Jun 29;9(50):29414-29430.] the proportion of uPAR-positive cells was described to be 64.6%, 59.1%, and 46.0%, respectively after one, two and three rounds of transfection using a canonical Wild-Type Cas9, demonstrating PLAUR to be a hard target to be knocked-out. Moreover, A375 and M6 are described by ATCC as hypotriploid cell lines so it is tough for the Cas9 to knockout selectively all the genomic copies of the PLAUR gene."

and

" Regarding the discrepancy in the lactate levels we know we will need to further investigate such phenomenon; however, we are confident that the different methods by which we modulated uPAR expression, here and in our previous work, might be responsible for the several differences observed. Indeed, using a siRNA we obtained an acute effect, downregulating uPAR expression for a maximum of 72h while exploiting the CRISPR/Cas9 we obtained a stable KO cell lines, forcing the cells to adapt to uPAR deprivation. Moreover, we are confident that loosing uPAR mRNA, all the miRNAs that are attached to its 3’ UTR, as reported by Li Santi et al. [Li Santi A, Gorrasi A, Alfieri M, Montuori N, Ragno P. A novel oncogenic role for urokinase receptor in leukemia cells: molecular sponge for oncosuppressor microRNAs. Oncotarget. 2018 Jun 12;9(45):27823-27834], might modulate several biological functions which will be the topic of our future investigations."

and

discussion about Akt and Erk signaling changes in discussion section.

The first and the second sentences were integrated in the Discussion section with the relative appropriated references. The phosphorylation levels of Akt and ERK were introduced and discussed in depth into the Discussion section.